

# High-throughput metabarcoding reveals the effect of physicochemical soil properties on soil and litter biodiversity and community turnover across Amazonia

Camila D. Ritter[1,2], Alexander Zizka[1,2], Fabian Roger[3], Hanna Tuomisto[4], Christopher Barnes[5], R. Henrik Nilsson[1,2,*] and Alexandre Antonelli[1,2,6,7,*]

[1] Gothenburg Global Biodiversity Centre, Göteborg, Sweden
[2] Department of Biological and Environmental Sciences, University of Gothenburg, Gothenburg, Sweden
[3] Centre for Environmental and Climate Research, Lund University, Lund, Sweden
[4] Department of Biology, University of Turku, Turku, Finland
[5] Natural History Museum of Denmark, University of Copenhagen, Denmark, Copenhagen, Denmark
[6] Gothenburg Botanical Garden, Göteborg, Sweden, Gothenburg, Sweden
[7] Department of Organismic and Evolutionary Biology, Harvard University, Cambridge, United States of America

[*] These authors contributed equally to this work.

Corresponding author
Camila D. Ritter, camila.ritter@gu.se, kmicaduarre@gmail.com

## ABSTRACT

**Background**. Knowledge on the globally outstanding Amazonian biodiversity and its environmental determinants stems almost exclusively from aboveground organisms, notably plants. In contrast, the environmental factors and habitat preferences that drive diversity patterns for micro-organisms in the ground remain elusive, despite the fact that micro-organisms constitute the overwhelming majority of life forms in any given location, in terms of both diversity and abundance. Here we address how the diversity and community turnover of operational taxonomic units (OTU) of organisms in soil and litter respond to soil physicochemical properties; whether OTU diversities and community composition in soil and litter are correlated with each other; and whether they respond in a similar way to soil properties.

**Methods**. We used recently inferred OTUs from high-throughput metabarcoding of the 16S (prokaryotes) and 18S (eukaryotes) genes to estimate OTU diversity (OTU richness and effective number of OTUs) and community composition for prokaryotes and eukaryotes in soil and litter across four localities in Brazilian Amazonia. All analyses were run separately for prokaryote and eukaryote OTUs, and for each group using both presence-absence and abundance data. Combining these with novel data on soil chemical and physical properties, we identify abiotic correlates of soil and litter organism diversity and community structure using regression, ordination, and variance partitioning analysis.

**Results**. Soil organic carbon content was the strongest factor explaining OTU diversity (negative correlation) and pH was the strongest factor explaining community turnover for prokaryotes and eukaryotes in both soil and litter. We found significant effects also for other soil variables, including both chemical and physical properties. The correlation between OTU diversity in litter and in soil was non-significant for eukaryotes and weak

for prokaryotes. The community compositions of both prokaryotes and eukaryotes were more separated among habitat types (terra-firme, várzea, igapó and campina) than between substrates (soil and litter).

**Discussion**. In spite of the limited sampling (four localities, 39 plots), our results provide a broad-scale view of the physical and chemical correlations of soil and litter biodiversity in a longitudinal transect across the world's largest rainforest. Our methods help to understand links between soil properties, OTU diversity patterns, and community composition and turnover. The lack of strong correlation between OTU diversity in litter and in soil suggests independence of diversity drives of these substrates and highlights the importance of including both measures in biodiversity assessments. Massive sequencing of soil and litter samples holds the potential to complement traditional biological inventories in advancing our understanding of the factors affecting tropical diversity.

## INTRODUCTION

Tropical rainforests are mega-diverse and environmentally heterogeneous biomes, and their biodiversity has been shown to vary considerably over space. In Amazonia, the world's largest rainforest that covers most of northern South America, geology and soil physicochemical properties are often considered crucial in regulating the biotic dynamics, vegetation, and diversity patterns at local to regional scales (*Vogel et al., 2009*; *Laurance et al., 2010*; *Higgins et al., 2011*; *Friesen et al., 2011*; *Tuomisto et al., 2016*).

For instance, diversity and community composition of plants are influenced by geology and physicochemical soil properties (e.g., *Vogel et al., 2009*; *Laurance et al., 2010*; *Higgins et al., 2011*; *Friesen et al., 2011*; *Tuomisto, Zuquim & Cárdenas, 2014*; *Tuomisto et al., 2016*; *Tedersoo et al., 2016*). In particular, the availability of soil nutrients and soil cation concentration are important factors determining plant species composition and turnover (*Tuomisto et al., 2003*; *Laurance et al., 2010*; *Higgins et al., 2011*; *Baldeck et al., 2016*; *Tuomisto et al., 2016*; *Cámara-Leret et al., 2017*). Additionally, soil properties, in particular phosphorus, can affect the taxonomic composition of microbial communities (*Buckley & Schmidt, 2001*; *Faoro et al., 2010*; *Navarrete et al., 2013*). In addition, pH is known to shape microbial diversity (e.g., *Osborne et al., 2011*; *Kuramae et al., 2012*; *Bates et al., 2013*; *Barnes et al., 2016*).

Different soil layers may show different patterns of biodiversity (*Hinsinger et al., 2009*). For instance, the taxonomic composition of nematode species clustered in six trophic guilds (bacterial feeders, fungal feeders, root associates, plant parasites, omnivores, and predators) has been found to vary between the mineral soil layer and the organic matter layer (litter) above it (*Porazinska et al., 2012*). In Amazonia, litter layers vary with habitat type and the length of the inundation period. Unflooded forests (terra-firme) are particularly rich in litter. In addition, flooded forests are also rich in litter and the litter layer increases with

increasing length of the inundation period (*Myster, 2017*). Besides inundation, several other factors influence litter accumulation and thereby decomposition rates and nutrient cycling. These include leaf abscission and species composition (e.g., *Gregorich et al., 2017*), which have implications for the diversity and community structure of soil and litter-inhabiting organisms. However, the strength of soil-litter interactions varies, and a study conducted in Canada reported no influence of soil physicochemical properties on litter decomposition (*Gregorich et al., 2017*). This may indicate a difference in biodiversity patterns (due to different drivers and biomass) between soil and litter layers.

The diversity and composition of Amazonian soil and litter communities remain poorly understood, despite recent studies on soil micro-organismic communities (e.g., *Basset et al., 2012*; *Mahé et al., 2017*; *Ritter et al., 2018*). This lack of knowledge, especially in taxonomic groups such as fungi, protists, nematodes, and bacteria, is problematic given the important roles of these groups in a wide range of biotic processes (*Falkowski, Fenchel & Delong, 2008*; *Stajich et al., 2009*; *Friesen et al., 2011*). To tackle his lack of knowledge, high-throughput amplicon-based sequencing analyses such as DNA metabarcoding (*Taberlet et al., 2012*) now allow examination of soil diversity patterns (*Bardgett & Van Der Putten, 2014*). However, most studies so far have been focused on one or a few taxonomic groups, which renders general conclusions on the effects of soil properties on biodiversity difficult (e.g., *Faoro et al., 2010*; *Laurance et al., 2010*; *Navarrete et al., 2013*; *Barnes et al., 2016*). Understanding microbial diversity and communities and their relation to soil physicochemical properties on a broad taxonomic scale is therefore crucial in any location, but particularly so in mega-diverse regions such as Amazonia to access general conclusion about the abiotic drivers of diversity.

In this study, we test the effect of physicochemical soil properties on soil and litter biodiversity and community turnover at four localities along a west-to-east transect across Brazilian Amazonia. We base diversity estimates on operational taxonomic units (OTUs) from environmental DNA of the ribosomal 16S (prokaryote) and nuclear ribosomal 18S (eukaryote) genes. Specifically, we seek to answer the following questions: Are OTU diversity and turnover related to physical and chemical soil properties? If so, what are the most important soil properties? Are OTU diversity and community composition correlated when quantified for the litter layer vs. the underlying soil? All questions are addressed separately for eukaryotes (18S) and prokaryotes (16S) using both presence-absence and abundance data.

## MATERIALS AND METHODS

### Sampling design and localities

We sampled four localities along the Amazon River (Fig. 1) following the sampling design of *Tedersoo et al. (2014)*. Detailed locality descriptions are available in *Ritter et al. (2018)*. Briefly, we sampled all depths of the litter layer above the mineral soil (all organic matter, including leaves, roots, and animal debris) and the top 5 cm of the mineral soil in a total of 39 circular plots, each with a radius of 28 m. We chose 20 random trees inside each plot and collected litter and soil on both sides of each tree. We then pooled the samples by substrate

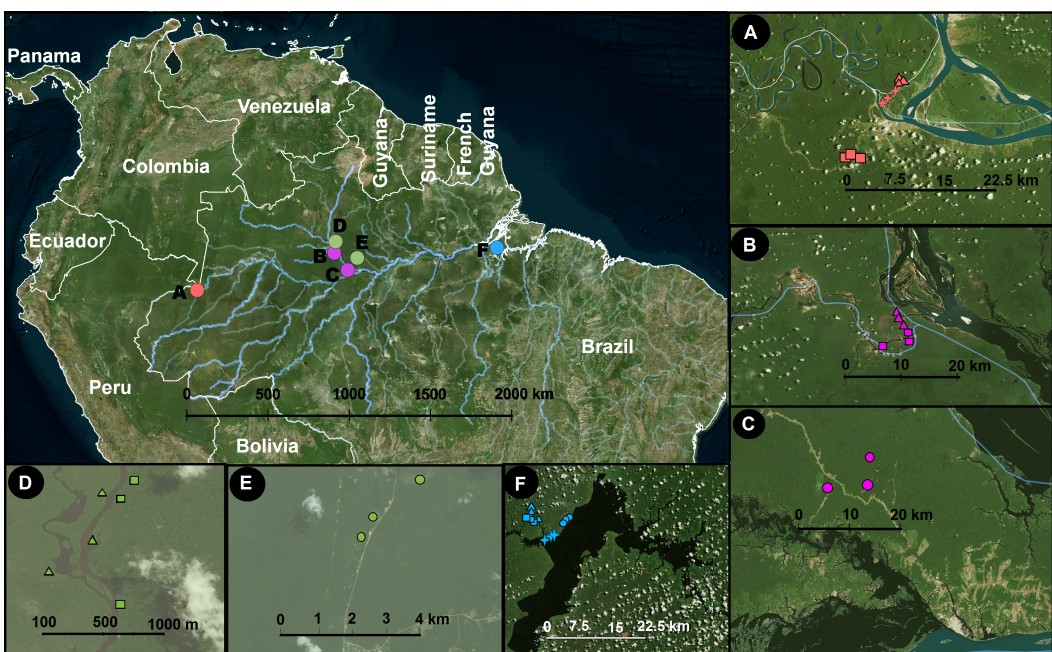

**Figure 1  Study area and sampling locations.** Inset panels show details of each locality. A, Benjamin Constant; B, Jaú; C, Jaú, naturally open areas; D, Cuieras; E, Cuieras, naturally open areas; and F, Caxiuanã. The symbols in A–F represent different vegetation types that are linked with different soil properties: circles, open areas; triangles, forest seasonally flooded by black water rivers; squares, unflooded forest; and crosses, forest seasonally flooded by white water rivers. The sampling strategy was designed to cover a wide longitudinal range in Amazonia. The map was constructed using *QGIS (2012)*.

to obtain one soil and one litter sample per plot. Each sample was stored in a plastic bag with the same weight than soil/litter samples of sterilized white silica gel of 1–4 mm grain size, pre-treated for two minutes of microwave heating (800 W) and 15 min of UV light. The bags were stored at room temperature (around 30 °C) in a dark box to avoid exposure to light. Once they arrived in Sweden, a period between 7–30 days, the samples were frozen (−20 °C). We sampled in different habitat types, which can be summarized as terra-firme, várzeas, igapós, and campinas. These are four of the commonly recognised main Amazonian environments. Terra-firme is characterised by not being inundated during the annual flood season, and terra-firme forests generally have tall stature and complex canopy structure (*IBGE, 2004*). In contrast, várzeas and igapós are seasonally flooded and remain submerged during parts of the year. Várzeas are flooded by white-water rivers, which carry a large load of suspended sediments, whereas igapós are flooded by black-water rivers, which bring a small load of suspended sediments but a high concentration of organic acids (*Junk et al., 2011*). Finally, campinas have nutrient-impoverished sandy soils and forests with a reduced stature and relatively simple canopy structure (*Prance, 1996*; *IBGE, 2004*).

Our sampling was carried out in four areas: *Benjamin Constant* (9 plots covering terra-firme, várzea and igapó), our westernmost locality, approximately 1,100 km west of Manaus in the upper Amazonas River (4.383°S, 70.017°W; Fig. 1A); *Jaú national park* (6 plots covering terra-firme and igapó; 1.850°S, 61.616°W; Fig. 1B) and *Novo Airão* (3

plots covering campinas; 2.620°S, 60.944°W; Fig. 1C), on the west side of the Negro River; *Reserva do Cuieras* (six plots covering terra-firme and igapó; 2.609°S, 60.217°W; Fig. 1D) and *Reserva da Campina* (three plots covering campinas; 2.592°S, 60.030°W; Fig. 1E), on the east side of the Negro River; and *Caxiaunã* (12 plots covering campinas, terra-firme, várzea, and igapó), a national forest located 350 km west of Belém (1.7352°S, 51,463°W; Fig. 1F), which constitutes our easternmost locality. The sample collection was authorized by Brazilian authorities: ICMBio (registration number 48185-2) and IBAMA (registration number 127341).

## Physicochemical soil analyses

We determined the physicochemical soil properties of each plot from three soil samples per plot, totalling 117 samples. The pH was measured in water (soil: water ratio 1:2.5). The exchangeable concentrations were measured for sodium (Na), potassium (K), and phosphorus (P) using Mehlich-1 extraction (unit $mg/dm^3$) and for calcium and magnesium (Ca, Mg) using KCl (1 mol/L) extraction (unit $cmol_c/dm^3$). The sum of all exchangeable bases (SB; which comprises $K^+$, $Ca^{2+}$, $Mg^{2+}$, and $Na^+$; unit $cmol_c/dm^3$) was then calculated. We also estimated exchangeable aluminium (Al and H+Al unit $cmol_c/dm^3$) extracted with calcium acetate (0.5 mol/L at pH 7.0), aluminium saturation index (m; unit %), and Base Saturation Index (V; unit %). The effective cation exchange capacity (t) as well as the cation exchange capacity (T) were measured at pH 7.0 (unit $cmol_c/dm^3$). The organic matter (O.M) was quantified (unit g/kg) and the organic carbon (C) was estimated from the organic matter as: C = O.M/1.724—Walkley-Black (unit g/kg). Soil texture was characterized by quantifying the fractions of clay (<0.002 mm), silt (0.002–0.05 mm), fine sand (0.05–0.2 mm), coarse sand (0.2–2 mm), and total sand (0.05–2 mm) (unit % of soil weight). We did not quantify nitrogen levels due to the highly volatile nature of nitrogen; its concentration changes quickly during sample storage due to the activity of soil bacteria, and freezing the samples in our remote sampling localities was not feasible. All analyses were commissioned from EMBRAPA Ocidental (Brazil), following the protocol described in *Donagema et al. (2011)*. Afterwards, we used the mean of the three soil samples from the same plot to obtain a representative value for the measurement of each variable for each plot.

## DNA extraction, amplification, and sequencing

The detailed laboratory procedures are described in *Ritter et al. (2018)*. Briefly, we extracted soil and litter using the PowerMax® Soil DNA Isolation Kit (MO BIO Laboratories, Carlsbad, CA, USA) following the manufacturer's instructions. The amplification of 16S was performed by Macrogen (Republic of Korea) following standard protocols, and sequencing was performed using the Illumina MiSeq 2×300 platform. For metabarcoding of the 18S gene, sequencing preparation was performed at the laboratory of the University of Gothenburg as described in *Ritter et al. (2018)* and the amplicons were sequenced at SciLifeLab (Stockholm, Sweden) using an Illumina MiSeq 2×250 machine.

## Sequence analyses

We used the USEARCH/UPARSE v9.0.2132 Illumina paired reads pipeline (*Edgar, 2013*) to filter out poor-quality sequences, de-replicate and sort reads by abundance, and remove

singletons. We inferred operational taxonomic units (OTU) at the 97% sequence similarity level as usually used for OTU clustering (meaning that sequences differing by more than 3% are considered to belong to different OTUs; *Stackebrandt and Goebel, 1994*; *Blaxter et al., 2005*). We used the SINA v1.2.10 for ARB SVN (revision 21,008; *Pruesse, Peplies & Glöckner, 2012*) taxonomic reference dataset for both markers and used SILVAngs 1.3 for taxonomic assignments (*Quast et al., 2012*).

## Construction of corrected OTU tables
### Presence/absence analyses
PCR biases, variation in the copy number of 16S/18S genes per cell/genome, as well as differences in size and biomass across the targeted organisms can compromise a straightforward interpretation of OTU reads as an abundance measure (*Elbrecht & Leese, 2015*; *Pawluczyk et al., 2015*). Since the number of observed OTUs is dependent on the number of reads, we first rarefied all samples to the lowest number of reads obtained from any one plot (22,209 for 16S and 25,144 for 18S; Fig. S1). One sample containing only 1,359 reads was excluded from the 18S data analysis prior to rarefaction to avoid having to downsize the other samples to such a low number of reads (*McMurdie & Holmes, 2014*). The OTU richness of each plot was computed after rarefaction using the function ''rarefy'' in the package vegan v. 2.4-3 (*Oksanen et al., 2007*) in R v3.3.2 (*R Development Core Team, 2017*). We subsequently transformed the rarefied OTU tables to presence/absence for both prokaryote (16S) and eukaryote (18S) data.

### Abundance analyses
Despite known limitations of methods, read abundances can be meaningful, especially for 16S. Therefore, we carried out analyses also using abundance data. We calculated true OTU diversity of order $q = 1$, which is equivalent to the exponential of the Shannon entropy (*Jost, 2006*). It can be interpreted as the effective number of OTUs, i.e., the number of OTUs in an idealised community where the geometric mean of the proportional OTU abundances is the same as in the original sample, but all OTUs are equally abundant (*Tuomisto, 2010*). The effective number of OTUs is more robust against biases arising from uneven sampling depth than the simple number of OTUs, so for diversity we used the unrarefied read counts as OTU abundance. However, the results were virtually identical when we used the rarefied OTU table (correlation = 1 for both 16S and 18S). For the remaining abundance-based analyses, we transformed read counts using the ''varianceStabilizingTransformation'' function in DESeq2 (*Love, Huber & Anders, 2014*) as suggested by *McMurdie & Holmes (2014)*. This transformation normalizes the count data with respect to sample size (number of reads in each sample) and variances, based on fitted dispersion-mean relationships (*Love, Huber & Anders, 2014*).

## Statistical analyses
### Preparation of environmental data
We first normalised all soil variables to zero mean and unit variance using the ''scale'' function of vegan. We then performed two principal component analyses (PCAs) to reduce the number of variables. The first PCA used the chemical soil properties, i.e., all

variables based on concentrations of elements. The second PCA used the physical soil properties, i.e., grain size classes. We input missing sand, silt, and clay information for three plots, based on regression weights from the observed data using the mice v. 2.30 R package (*Buuren & Groothuis-Oudshoorn, 2011*) before performing the PCAs. We used the first axis of each PCA (explaining 66% and 65% of the total variation, respectively) in the subsequent analyses. Given the expected importance of soil organic carbon content (*Nielsen et al., 2011*) and pH (*Lauber et al., 2009*), we used these as independent variables.

### Hypothesis testing

For all analyses we used pH, organic carbon, chemical PC1, and physical PC1 as explanatory variables. All analyses were carried out using presence-absence data and relative abundance data in parallel: in the case of diversity, richness corresponds to presence-absence data and effective number of OTUs to relative abundance data. Overall, each kind of analysis was carried out eight times: one for each of the eight possible combinations of organism group (prokaryote or 16S, eukaryote or 18S), substrate (soil, litter) and abundance measure (presence-absence, proportional abundance).

### Do OTU diversities reflect physical or chemical soil properties?

To address the first question, we performed Bayesian general linear models (GLM), as implemented in the R-INLA v. 17.6.20 R package (*Rue et al., 2009*). The response variables were the eight different variants of OTU diversity in turn, and in each case the soil properties were used as explanatory variables. We tested the effect of spatial auto-correlation by comparing analyses of standard GLMs with GLM analysis using stochastic partial differential equations (SPDE) that explicitly consider spatial correlation.

### Do OTU community turnovers reflect differences in physical and chemical soil properties?

To address the second question, we performed multiple regressions on dissimilarity matrices (MRM), as implemented in the function "MRM" of the R package *ecodist* v.2.0.1 (*Goslee & Urban, 2007*). The response variables were dissimilarity matrices based on the eight different variants of OTU turnover (as calculated using the Jaccard dissimilarity) in turn. In each case, the explanatory variables were four distance matrices based on soil properties and one geographical distance matrix (all calculated using Euclidean distances). Statistical significance of the regression coefficients was determined with 10,000 permutations. Additionally, we used variance partitioning analysis based in dissimilarity matrixes to quantify the unique and shared contributions of each of the explanatory variables to explaining variance in the response distance matrix (*Tuomisto & Ruokolainen, 2006*).

### Does OTU diversity of the litter layer predict the OTU diversity of the underlying soil?

To address the third question, we analysed the relationship between litter and soil OTU richness and diversity for prokaryotes (16S) and eukaryotes (18S) using a linear regression model (the *lm* function in R).

***Are OTU community turnover patterns in the litter layer similar to those in the underlying soil?***

To address the fourth question, we first performed non-metric multidimensional scaling (NMDS) ordinations as implemented in the metaMDS function in the R package vegan. Compositional dissimilarity was quantified with the Jaccard dissimilarity index. Ordinations based on the same organism group and abundance data type but different substrates were then compared. Next, we used the Permutational Analysis of Variance (PERMANOVA) to assess whether substrate type has an effect on community composition. Finally, we illustrated which were the dominant taxonomic groups (phyla or kingdom) with bar-plots. As with the other questions, all analyses were repeated for all possible combinations of organism group (prokaryotes or 16S and eukaryotes or 18S) and abundance data type (presence-absence and proportional abundance, in the bar-plots we used the rarefied abundance data).

Additional R packages we used for data curation and visualization were tidyverse v. 1.1.1 (*Wickham, 2017*), Hmisc v. 4.0-3 (*Harrell Jr & Dupont, 2008*), ggfortify v. 0.1.0 (*Tang, Horikoshi & Li, 2016*), gridExtra v. 2.2.1 (*Auguie, Antonov & Auguie, 2016*), ggplot2 (*Wickham, 2016*), entropart (*Marcon & Hérault, 2015*), broom v.0.4.4 (*Robinson, 2017*), and viridis v. 0.4.0 (*Garnier, 2016*). Scripts for all analyses are provided in the supplementary material.

## RESULTS

### OTU diversity and turnover in relation to soil properties (research 1 and 2)

In the physical soil data PCA, large values on the first PC were associated with coarse texture (coarse sand fraction loading 0.52, total sand fraction loading 0.55) and small values with fine texture (silt loading $-0.45$, clay loading $-0.38$; Table S2). The flooded forests (igapós and várzeas) generally had fine-textured soils (negative values of PC1). The unflooded forests (terra-firme and campinas) were more widely distributed along PC1, with some plots having similar values with várzeas and igapós (Fig. 2A). In the chemical soil data PCA, large values on the first PC were associated with poor soils. The most negative loading was $-0.35$ for the sum of exchangeable bases (SB), and the largest positive loading was 0.29 for aluminium saturation index (Table S3). The habitat types were not well separated along the first axis of the chemical PCA, as most plots of all habitat types had poor soils (large values of PC1) and just a few scattered várzea and igapó plots had more cation-rich soils (small values of PC1; Fig. 2B).

In general, organic carbon and pH had the strongest effects on OTU diversity. This was the case both for prokaryotes and eukaryotes, for richness and effective number of OTUs and for soil and litter (Table 1). In addition, PC1 of chemical soil properties was an important predictor for prokaryotic OTU richness in the soil and litter, with stronger effect in soil than in litter, and for prokaryotic effective number of OTUs just in soil. For eukaryotes, soil texture had an important effect on OTU diversity, albeit different on each substrate: positive for soil and negative for litter (Table 1). Overall, soil properties had strong effects on OTU diversity in litter.

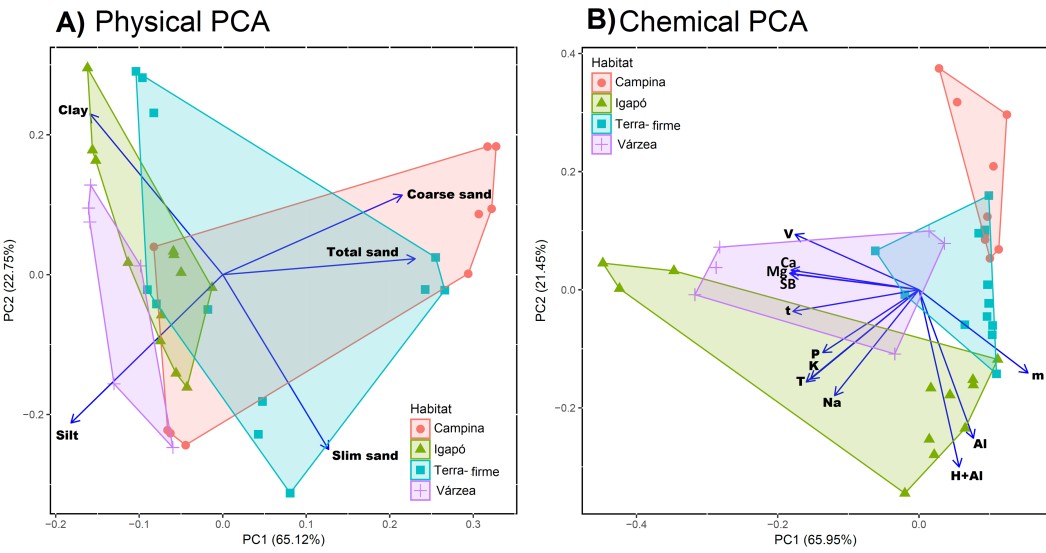

**Figure 2  Physical and chemical soil similarity of sample sites across Amazonia.** The figure shows the study sites—coloured by habitat type—on the first two axes of a Principle Component Analysis for (A) physical properties (silt, clay and fine, coarse, and total sand fraction) and (B) chemical proprieties (phosphorus (P), exchangeable bases (Na, K, Ca, and Mg), the sum of all exchangeable bases (SB), exchangeable aluminium (Al and H+Al), saturation index by aluminium (m), base saturation index (V), effective cation exchange capacity (t), and cation exchange capacity (T)). The blue rows show the values of each variable loadings in the two firs PCs. For physical PCA we can observe that flooded forest (igapós and várzeas) are associated with negative values in the first PC axis and a more spread distribution of terra-firme and campinas. For chemical PCA there is no separation of flooded forest, but campinas group in most positive values in the first PCA axis followed by terra-firmes.

OTU community turnover was significantly associated with soil properties, especially with organic carbon and pH, which were significant for all communities. The pH effect was strong for all prokaryote (16S) datasets and for eukaryotes (18S) in soil when relative abundance data were used (Table 2). Organic carbon had the strongest effect for eukaryotes in soil when presence/absence data were used and for eukaryotes in litter with both presence/absence and relative abundance data. Chemical PC1 was significant for prokaryotes in soil (both presence/absence and relative abundance) and for eukaryotes in litter when presence/absence data were used. Texture PC1 was significant only for eukaryotes in litter (both presence/absence and relative abundance; Table 2). Geographical distance was a significant explanatory factor for all datasets, but as closer places usually are more environmentally similar, we cannot separate the effect of spatial correlation from soil property effects.

A moderate percentage of the variation in Jaccard dissimilarities was explained by soil physicochemical properties in the presence/absence data, for prokaryotes (31% in soil and 35% in litter; Fig. 3). For eukaryotes, the total explanatory power of soil physicochemical properties was smaller (12% in soil and 16% in litter). For prokaryotes and eukaryotes, the litter communities were more structured by soil characteristics than were the soil communities. All variables explained small but significant proportions of the variance

**Table 1 Soil effects on OTU richness and Shannon diversity.** The importance of soil properties differed between taxon, substrate and diversity metrics. Carbon content and pH were important in most of the cases. The table shows the coefficients of each predictor in four Bayesian general multivariate regression model using Stochastic Partial Differential Equations (SPDE) that explicitly consider spatial correlation, modelling OTU richness and effective number of OTUs dependent on soil properties for eukaryotes and prokaryotes in litter and soil, respectively. As the organic carbon content and pH are important variables for soil biota, we use them as independent variables. Bold indicates important predictor variables (credible intervals not crossing zero).

| Taxon | Substrate | Predictor | OTU richness | | | Effective number of OTUs | | |
|---|---|---|---|---|---|---|---|---|
| | | | Mean | 0.025 quantile | 0.975 quantile | Mean | 0.025 quantile | 0.975 quantile |
| Prokaryotes | Soil | Intercept | 6.14 | 2.31 | 9.95 | 6.07 | 2.03 | 10.43 |
| | | pH | **0.22** | **0.16** | **0.27** | **0.22** | **0.17** | **0.27** |
| | | Carbon | **−0.13** | **−0.19** | **−0.08** | **−0.15** | **−0.20** | **−0.10** |
| | | Chemical | **−0.08** | **−0.14** | **−0.02** | **−0.08** | **−0.14** | **−0.02** |
| | | Physical | **0.02** | **0.00** | **0.04** | 0.02 | −0.01 | 0.04 |
| | Litter | Intercept | 4.14 | −4.78 | 12.65 | 5.04 | −3.97 | 13.71 |
| | | pH | 0.03 | −0.03 | 0.08 | **0.05** | **0.00** | **0.10** |
| | | Carbon | **−0.23** | **−0.28** | **−0.17** | **−0.21** | **−0.26** | **−0.16** |
| | | Chemical | **−0.07** | **−0.12** | **−0.01** | −0.05 | −0.11 | 0.01 |
| | | Physical | **0.13** | **0.10** | **0.15** | **0.12** | **0.10** | **0.15** |
| Eukaryotes | Soil | Intercept | 3.28 | −7.36 | 13.39 | 3.22 | −6.49 | 13.22 |
| | | pH | **0.30** | **0.22** | **0.38** | **0.33** | **0.25** | **0.41** |
| | | Carbon | **−0.36** | **−0.46** | **−0.26** | **−0.36** | **−0.46** | **−0.27** |
| | | Chemical | −0.08 | −0.20 | 0.04 | −0.05 | −0.17 | 0.06 |
| | | Physical | **0.11** | **0.07** | **0.15** | **0.10** | **0.07** | **0.14** |
| | Litter | Intercept | 4.93 | −10.81 | 19.82 | 6.07 | −10.14 | 21.34 |
| | | pH | **0.24** | **0.15** | **0.34** | **0.26** | **0.16** | **0.35** |
| | | Carbon | **−0.22** | **−0.32** | **−0.13** | **−0.23** | **−0.32** | **−0.14** |
| | | Chemical | −0.01 | −0.12 | 0.10 | 0.00 | −0.10 | 0.11 |
| | | Physical | **−0.14** | **−0.19** | **−0.09** | **−0.15** | **−0.20** | **−0.10** |

in all communities and showed some weak but significant interactions considering presence/absence matrices (Fig. 3) and a similar, strong proportion of the variance in abundance data (Fig. S2). Organic carbon had the strongest effect in all substrates and for both organism groups (ranging from 0.03 for eukaryotes in soil through 0.05 for eukaryotes in litter to 0.08 for prokaryotes in both soil and litter).

## Similarities in OTU diversity and turnover patterns between litter and soil (research 3–4)

We found a weak positive regression between OTU richness of prokaryotes in litter and in soil (adj. $R^2 = 0.25$, $p < 0.001$; Fig. 4A) and between the effective number of prokaryote OTUs in litter and in soil (adj. $R^2 = 0.1$, $p = 0.03$; Fig. 4B). For eukaryotes, the corresponding correlations were not significant (Figs. 4C and 4D). The plot "CXNCAMP3" had very low soil OTU richness, and excluding this data point strengthened the correlation of OTU richness between soil and litter for prokaryotes (to adj. $R^2 = 0.46$, $p < 0.001$; Fig. S3A), but not for eukaryotes (Fig. S3B).
**Table 2  Association between environmental distance and community turnover.** Community dissim-ilarity is significantly associated with geographical and soil environmental distance for eukaryote and prokaryote communities in soil and litter. The Multiple Regressions were based on the geographical dis-tance, Euclidean distance matrices of soil properties and community Jaccard dissimilarity indexes. Geo-graphic distances were significant for all communities turnover; however, as geographical closest places are usually more environmental similar, we cannot separate the effect of soil properties from the spatial corre-lation. All community turnovers were significant with 10,000 permutations ($p < 0.001$) with the follow $R^2$: prokaryote soil ($R^2 = 0.36$ for presence/absence and $R^2 = 0.36$ for relative abundance), prokaryote litter ($R^2 = 0.39$ for presence/absence and $R^2 = 0.35$ for relative abundance), eukaryote soil ($R^2 = 0.21$ for pres-ence/absence and $R^2 = 0.20$ for relative abundance) and eukaryote litter ($R^2 = 0.32$ for presence/absence and $R^2 = 0.30$ for relative abundance).

| Taxon | Substrate | Predictor | Presence/absence | | Relative abundance | |
| --- | --- | --- | --- | --- | --- | --- |
| | | | Coefficients | *p* value | Coefficients | *p* value |
| Prokaryotes | Soil | Intercept | −24.28 | 1.00 | −8.63 | 1.00 |
| | | Geo_dist | **0.15** | **0.00** | **0.15** | **0.00** |
| | | Chemical | **0.23** | **0.00** | **0.21** | **0.01** |
| | | Physical | **0.18** | **0.01** | **0.17** | **0.01** |
| | | pH | **0.30** | **0.00** | **0.34** | **0.00** |
| | | Carbon | **0.20** | **0.01** | **0.16** | **0.03** |
| | Litter | Intercept | −60.59 | 1.00 | −37.45 | 1.00 |
| | | Geo_dist | **0.19** | **0.00** | **0.20** | **0.00** |
| | | Chemical | **0.18** | **0.00** | 0.11 | 0.09 |
| | | Physical | **0.16** | **0.00** | **0.19** | **0.00** |
| | | pH | **0.32** | **0.00** | **0.33** | **0.00** |
| | | Carbon | **0.32** | **0.00** | **0.28** | **0.00** |
| Eukaryotes | Soil | Intercept | 43.54 | 1.00 | 59.63 | 1.00 |
| | | Geo_dist | 0.09 | 0.06 | **0.10** | **0.03** |
| | | Chemical | 0.12 | 0.16 | 0.11 | 0.21 |
| | | Physical | **0.15** | **0.04** | **0.16** | **0.03** |
| | | pH | **0.22** | **0.00** | **0.24** | **0.00** |
| | | Carbon | **0.29** | **0.00** | **0.23** | **0.01** |
| | Litter | Intercept | −8.86 | 1.00 | 10.65 | 1.00 |
| | | Geo_dist | **0.15** | **0.00** | **0.17** | **0.00** |
| | | Chemical | **0.26** | **0.00** | **0.22** | **0.00** |
| | | Physical | 0.09 | 0.13 | 0.05 | 0.40 |
| | | pH | **0.19** | **0.00** | **0.24** | **0.00** |
| | | Carbon | **0.33** | **0.00** | **0.29** | **0.00** |

The OTU communities in litter and in soil tended to be separated in the NMDS ordination space, although there was some overlap especially for the igapó plots (Fig. 5). The PERMANOVA test indicated weak but significant effects (all $p < 0.001$) of substrate type on compositional dissimilarities of both prokaryotes ($R^2 = 0.06$, $F = 5.83$, for presence/absence data and $R^2 = 0.07$, $F = 6.7$, for abundance data) and eukaryotes ($R^2 = 0.03$, $F = 2.8$ for presence/absence data and $R^2 = 0.04$, $F = 3.74$, for abundance data). Habitat type had an even stronger effect on the compositional dissimilarities of both prokaryotes ($R^2 = 0.17$, $F = 5.48$, for presence/absence data and $R^2 = 0.18$, $F = 5.68$, for abundance data) and eukaryotes ($R^2 = 0.1$, $F = 2.8$, for presence/absence data and $R^2 = 0.1$,

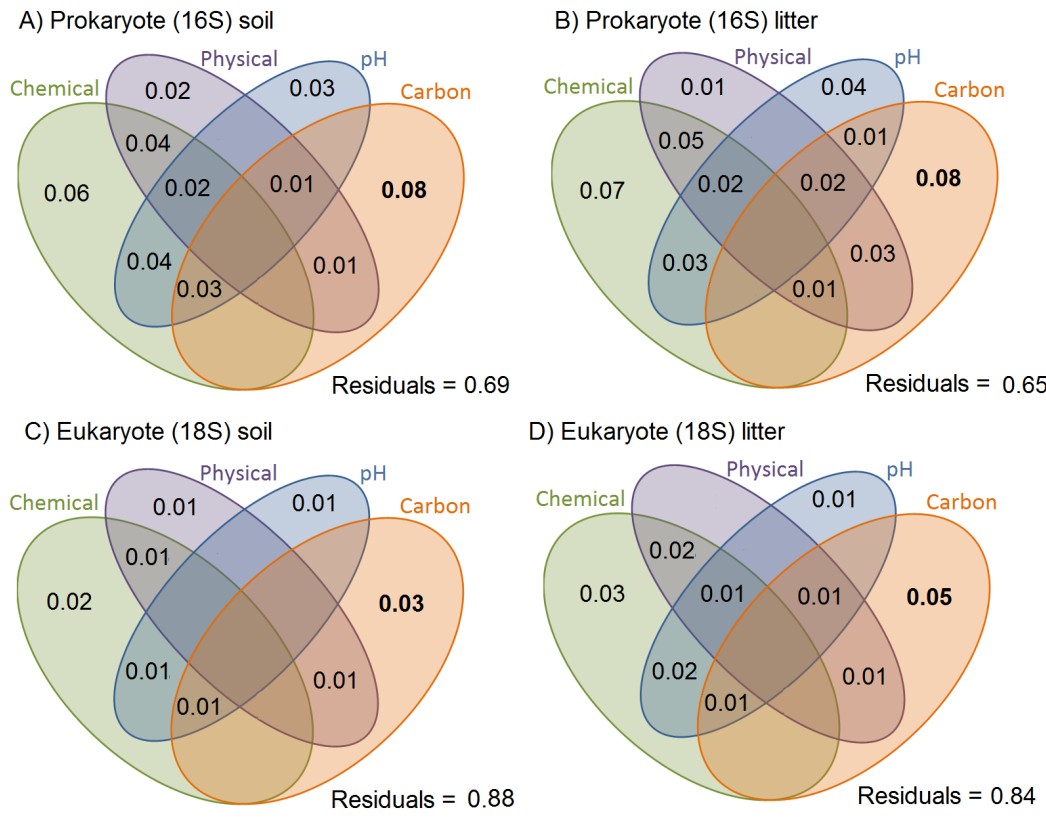

**Figure 3** **Variation in OTU community composition in Amazonian soil samples explained by soil characteristics.** Results of the variance partition analysis based on Jaccard dissimilarity distance-based analysis. Small but significant proportions of soil and litter communities vary with soil variables, and a small but significant proportion shows variation shared by soil variables. All values represent the proportion of variation explained by the factor/interaction. Chemical variables are shown in green (based on the first PCA of chemical variables, see Table S3 for details), physical variables in purple (based on first PCA axis of soil texture, see Table S2 for details), pH in blue, and carbon content in orange. The prokaryote communities are more structured by soil characteristics than are the eukaryote ones. Inside each taxonomic group, the litter communities are more structured by soil characteristics than are the soil communities.

$F = 2.98$, for abundance data). Taxonomic composition at the phylum and kingdom (for fungi) level was similar in litter and in soil both for prokaryotes and for eukaryotes (Fig. 6). However, in prokaryotes Actinobacteria is the second most abundant phylum in litter when taking into account the relative abundances (Fig. 6).

## DISCUSSION

### Soil predictors of OTU diversity and community turnover

In this study, we tested the impact of physicochemical soil properties on the OTU diversity (richness and effective number of OTUs) and community turnover of prokaryotes and eukaryotes in soil and litter across Brazilian Amazonia. We found that that the soil properties we quantified had variable effects on OTU diversity and community turnover for litter and soil, and the effect varied between prokaryotic and eukaryotic organisms. The variable

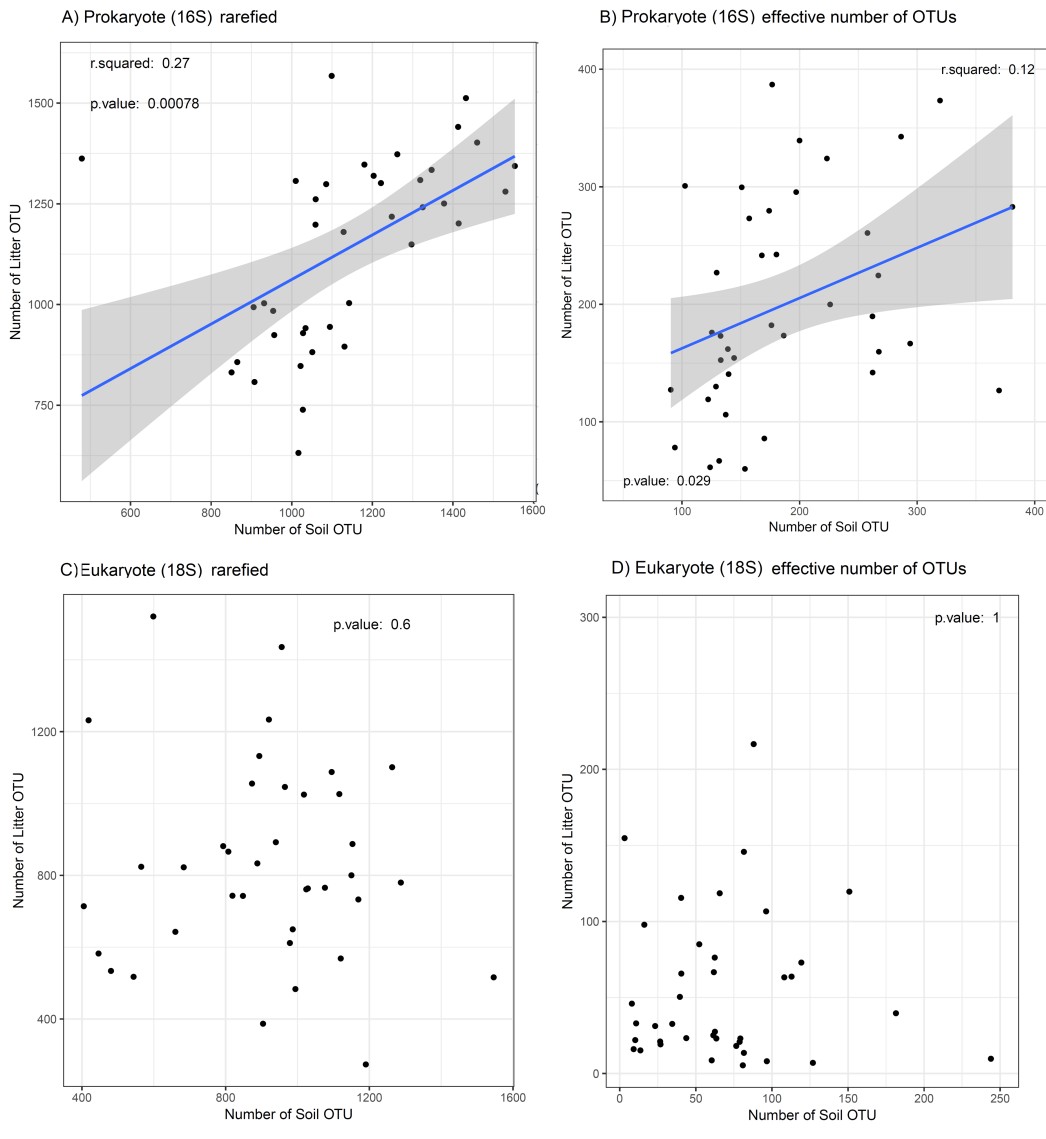

**Figure 4 Relation between OTU diversity in soil and litter.** Prokaryotes (16S) are showed in (A) OTU richness and (B) effective number of OTUs; eukaryotes (18S) are showed in (C) OTU richness and (D) effective number of OTUs in the Amazonian soil samples. The blue line shows a linear regression with standard error indicated by the shaded area for significant correlations. The relationship between soil layers (litter *vs* soil) is weak and differs between taxa, with only prokaryotes showing a significant correlation for richness and effective number of OTUs. This result suggests that it OTU diversity in litter is unsuitable as proxy for the OTU diversity in the soil and vice versa.

with the highest explanatory power was overall organic carbon for both prokaryotes and eukaryotes. OTU diversity and community turnover were better explained by soil properties in litter than in soil.

Considering the results from the linear models, in general organic carbon and pH were the strongest factors in explaining soil prokaryotic and litter and soil eukaryotic diversity. Our results show a positive correlation between soil pH and OTU diversity, which is

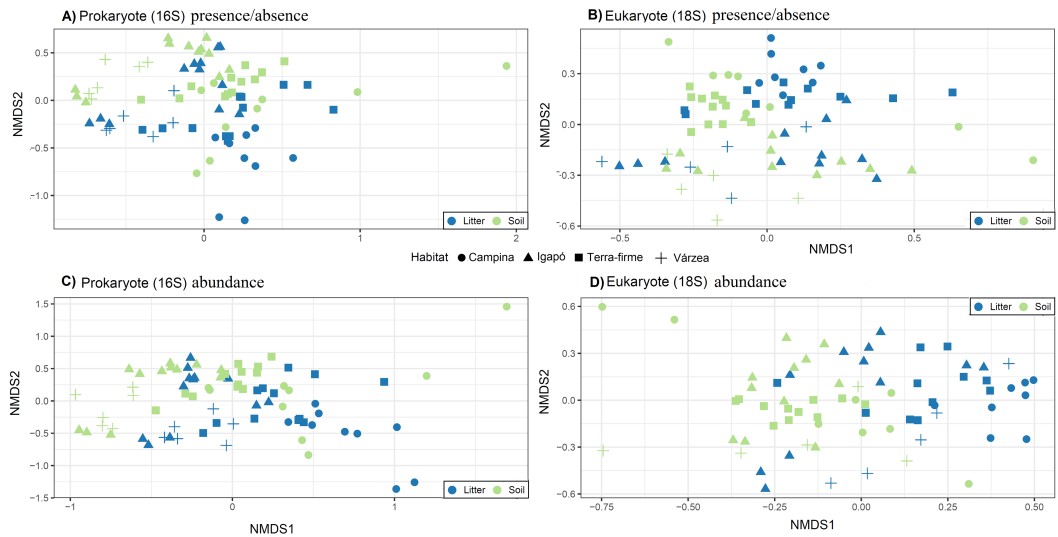

**Figure 5 Community structure related to substrate type (litter and soil) and habitat types.** Visualization of differences in OTU composition (measured with the presence/absence matrixes using Jaccard dissimilarity index in A and B; and measured with relative abundance matrixes using Bray-Curtis dissimilarity index) using non-metric multidimensional scaling (NMDS) for (A) and (C) prokaryotes (16S) and (B) and (D) eukaryotes (18S). Symbols represent different habitats. Blue represent litter samples and green soil samples. A small but statistically significant (PERMANOVA test) separation between the substrates can be observed along the second ordination axis for both groups of organisms. The strongest and most significant separation is observed between habitat types.

expected since much of the soils in Amazonia are acidic. For instance, for soil samples, *Lauber et al. (2009)* found pH to be the main factor in explaining bacterial phylogenetic diversity and phylogenetic composition, where soils with pH between 4.5 and 8 had the highest bacterial diversity. Tropical forests with high macro-organismic diversity had soil with pH < 4.5 and had the lowest bacterial diversity (*Lauber et al., 2009*). In our samples, pH was overall low and its variation was moderate, from 3.65 to 5.14, thereby in less acid soil we found highest OTU diversity considering both richness and effective number of OTUs.

We also found that variation in pH was significant for all community turnovers, and that it was the strongest variable in explaining community turnover for prokaryotes (in both soil and litter) and for eukaryotes in soil (Table 2). However, we found that pH had no effect on prokaryote richness in litter (Table 1). The consistent effect of pH for prokaryotes and eukaryotes in the soil, with significant effect in community turnover and diversity, but inconsistent effect in litter support the findings of *Gregorich et al. (2017)*, who found no correlation of soil properties and litter decomposition. This points to the independence of the environmental factors regulating each substrate.

We found significant effects of variation in organic carbon on all community turnovers with the strongest effect for eukaryotes in litter (Table 2). Additionally, we found a negative correlation of soil organic carbon with OTU diversity for all groups (Table 1). Soil biodiversity has previously been found to have an effect on carbon sequestration (*Wagg*

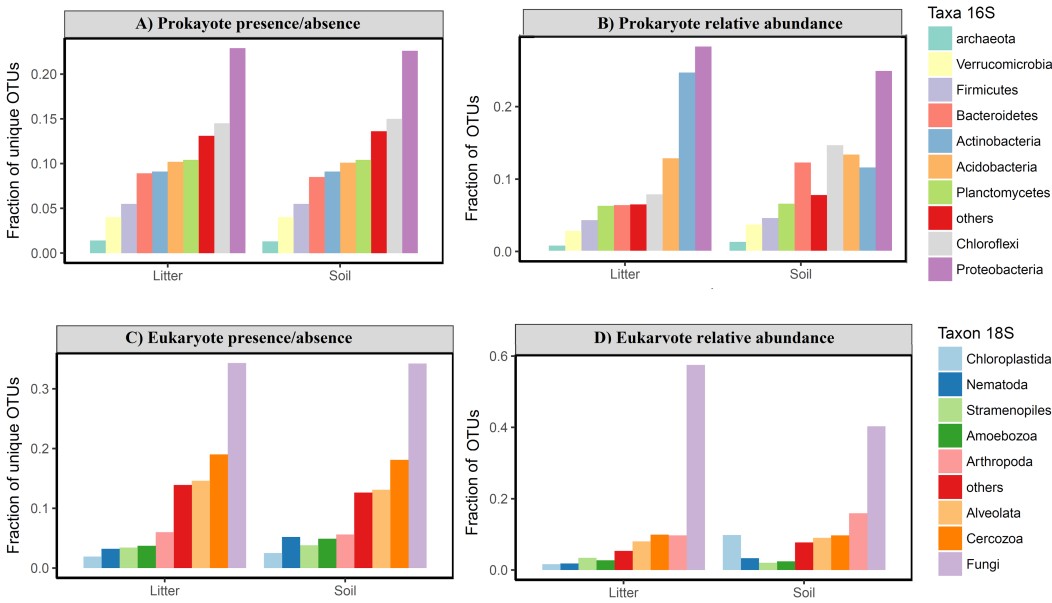

**Figure 6** **Taxonomic composition of Amazonian soil and litter micro-organismic communities.** The plots show the fraction of OTUs divided by taxonomic group for (A) relative frequency of OTU presence for prokaryotes; (B) relative abundance of OTU for prokaryotes; (C) relative frequency of OTU presence for eukaryotes; (D) relative abundance of OTU for eukaryotes. There is no clear taxonomic variation among groups in litter *vs* soil, in either the prokaryote or the eukaryote data for presence/absence. The relative abundance data shows a higher frequency of Actinobacteria in litter compared with soil and a higher abundance of Chloroflexi and Bacterioides in soil sample for prokaryotes. For eukaryotes is possible observe a highest relative abundance of Arthropoda and Chloroplastida in soil than litter samples.

*et al., 2014*). However, the relationship between soil biodiversity and carbon has varied across studies (*Nielsen et al., 2011*). Furthermore, *Fierer et al. (2012)* and *De Lima Brossi et al. (2014)* found that soil organic matter was related to microbial community composition in several different vegetation types. The negative correlation between soil organic carbon content and OTU diversity reported here might be related to high nutrient turnover in high-diversity soil/litter, keeping the carbon stock locked in aboveground biomass. Our results support the findings of *Wall et al. (2008)*, who found a positive influence of the richness of soil biota on decomposition rates in wet tropical environments. Along the same line, *Wagg et al. (2014)* found that soil diversity and soil community composition are related through nutrient cycling. Decreases in soil diversity and the related changes in community composition alter the communities' capacity to break down organic matter and recycle nutrients, slowing down the return of nutrients to the above-ground communities (*Wardle et al., 2004*). These findings stress the complex nature of carbon-diversity dynamics and the plant–soil feedback loop mediated by soil biota (*Mangan et al., 2010*). They furthermore highlight a connection between decomposition rates and biodiversity in Amazonia that should be better explored.

We found that prokaryotic community turnover is more strongly related with environmental distance than eukaryotic community turnover which is mostly dominated by fungi (fungi correspond to 35% of OTU richness and 50% of OTU relative abundance

in our eukaryotic data; Figs. 6C and 6D). This is in agreement with the results from a global bacterial and fungi soil sampling (*Bahram et al., 2018*). In our results, environmental distances explained only a limited percentage of OTU community turnover (31–35% for prokaryotes and 12–16% for eukaryotes; Fig. 3). This suggests that other factors, such as precipitation and bacterial-fungal antagonistic interactions (*Bahram et al., 2018*), also need to be considered to better understand the community turnover these organisms.

Biotic and abiotic interactions jointly determine soil properties, making it important to consider environmental and biological interactions between variables. Indeed, our variance analysis reveals several co-variances between soil properties, such as pH and organic carbon and physical and chemical properties. Although these interactions were weak, this analysis is important for providing a better understanding of the study system. Considering physicochemical soil properties, we had a partial separation of the major environmental types by the properties of their soils. It is in agreement with previous studies, which report an association of soil types and habitats in Amazonia (e.g., *Falesi, 1984*; *Prance, 1996*). The soil texture (first axis of the physical PCA) was well separated by the habitat types of flooded forests (igapós and várzeas), whereas terra-firme and campinas were more spread in physical properties. On the other hand, the first axis of chemical PCA was less well separated for flooded areas (igapós and várzeas). This result was expected since there is a variation within habitat types, especially flooded forests (*Kalliola et al., 1993*; *Tuomisto, Ruokolainen & Yli-Halla, 2003*; *Tuomisto, Zuquim & Cárdenas, 2014*; *Tuomisto et al., 2016*). Furthermore, terra-firme forests have been reported to vary in soil nutrients (e.g., *Tuomisto, Ruokolainen & Yli-Halla, 2003*; *Tuomisto, Zuquim & Cárdenas, 2014*; *Tuomisto et al., 2016*; *Fine et al., 2005*), consistent with the variation observed in our plots. However, due to the limited sampling in our studies, the variation we detected was small (Fig. 2B). Our finding that soil texture is similar among the flooded environments (várzeas and igapós) and that soil texture was an important factor for eukaryote diversity (both soil and litter) is consistent with the previously reported community similarity among these environmental types based on the data from the same samples (*Ritter et al., 2018*).

## Contrasting litter and soil diversity

The correlations between soil and litter OTU diversity (richness and the effective number of OTUs) were significant for prokaryotes but not for eukaryotes. This is congruent with previous reports that showed the independence of litter accumulation from properties of the underlying soils (*Gregorich et al., 2017*).

We expected a difference in taxonomic composition between litter and soil communities, with microbes dominating the soil (e.g., *Bates et al., 2013*; *Mahé et al., 2017*) and plant and nematode OTUs dominating the litter due to it mainly being composed of leaves and roots. However, we found the highest plant (Chloroplastida) richness and abundance in the soil samples. Furthermore, unlike *Porazinska et al. (2012)* who found a dominance of nematodes in the litter of tropical forests, we found very similar proportions of nematode OTUs in soil and litter, with the highest richness and abundance in the soil (Figs. 6C and 6D). This pattern of no clear taxonomic differences between soil and litter layers is

consistent with respect to all dominant groups at the phylum and kingdom levels (Fig. 6). This suggests that, on the Amazon basin scale, the taxonomic composition at higher levels (phylum and kingdom) is consistent between litter and soil with some variation in abundance for some groups, such as Arthropoda and Chloroplastida for eukaryotes and Actinobacteria, Bacterioides and Chloroflexi for prokaryotes (Fig. 6).

Interestingly, the prokaryotic phyla that dominated our samples were only partly the same as those found dominant in a large global dataset (Figs. 6A and 6B; *Delgado-Baquerizo et al., 2018*). While we also found Proteobacteria to be the most frequent phylum considering both presence/absence and relative abundance, the second most frequent phylum was Chloroflexi in presence/absence data for soil and litter and abundance for soil in our samples, while this Chloroflexi was only the 5th most abundant in the global dataset. Actinobacteria, the second most abundant phylum in the global database, was in our data the second most abundant phylum just for abundance in litter samples. Moreover, the rank-abundance distribution of the most dominant phyla was more even in our tropical sample than in the global sample, with Proteobacteria accounting for just over 20% of all reads (versus almost 40% in the global dataset) and eight phyla representing more than 5% of relative frequency each (>70% of relative frequency) versus only four phyla in *Delgado-Baquerizo et al. (2018)*. Taken together, these differences highlight the need for more studies across the Amazon basin to better characterize the taxonomic composition.

The OTU community compositions of both prokaryotes and eukaryotes were better explained by habitat type (terra firme, várzea, igapó, campina) than they were by substrate type (soil, litter), which was expected since both substrates should share a large number of organisms. The substrate types were weakly differentiated at the OTU level, but we could not observe any difference at the phylum or kingdom levels for presence-absence and only a small difference for abundance data (Fig. 6). For instance, fungi usually dominate eukaryotic soil communities in any environment, including tropical forests (*Tedersoo et al., 2017*), but the dominant fungal taxa (OTU) may vary considerably even at local and sub-local scales (*Urbanová, Šnajdr & Baldrian, 2015*). In a study conducted in the western parts of the Czech Republic, similar results for bacteria and fungi were found: the phylum level indicated the same taxonomic groups as dominant in soils and litter, but there were striking differences at the OTU level in these substrates (*Urbanová, Šnajdr & Baldrian, 2015*).

## CONCLUSIONS

In this study, we found OTU diversities to be related between soil and litter in prokaryotes, but not in eukaryotes. We also found that physicochemical soil properties can predict soil and litter diversity in Amazonia to some extent. In particular, we found a positive correlation for pH and a negative correlation for soil organic carbon content with respect to prokaryotic and eukaryotic OTU diversity. Furthermore, we found a significant effect of variation in soil organic carbon content on community turnover. In general, our results stress the complexity of soil-biodiversity relationships, and hence the importance of considering multiple factors and their interactions in the characterization of biodiversity patterns. Soil biodiversity is crucial for carbon cycling in terrestrial ecosystems, and our results suggest

that additional studies to better understand the relationship between diversity (above and belowground) and carbon cycles may help modelling carbon deposition and biodiversity patterns.

## ACKNOWLEDGEMENTS

We thank Anna Ansebo, Sven Toresson, and Ylva Heed for laboratory and administrative assistance; Mats Töpel for help with bioinformatics; and Quiterie Haenel for help with the DNA extractions and fruitful discussions. We thank Hans ter Steege for advice on sampling localities and experimental design. We are grateful to members of our research group for discussions and suggestions. Additional computational analyses were run at the University of Gothenburg bioinformatics cluster at the Department of Biological and Environmental Sciences (http://albiorix.bioenv.gu.se/).

### Funding

This study had primary financial support from CNPq (Conselho Nacional de Desenvolvimento Científico e Tecnológico - Brazil: 249064/2013-8) and the Swedish Research Council (B0569601). Antonelli is further supported by the European Research Council under the European Union's Seventh Framework Programme (FP/2007-2013, ERC Grant Agreement n. 331024), the Swedish Foundation for Strategic Research, a Wallenberg Academy Fellowship, the Faculty of Sciences at the University of Gothenburg, and the David Rockefeller Center for Latin American Studies at Harvard University. There was no additional external funding received for this study. The funders had no role in study design, data collection and analysis, decision to publish, or preparation of the manuscript.

### Grant Disclosures

The following grant information was disclosed by the authors:
CNPq (Conselho Nacional de Desenvolvimento Científico e Tecnológico - Brazil): 249064/2013-8.
Swedish Research Council: B0569601.
European Research Council: 331024.
Swedish Foundation for Strategic Research.
David Rockefeller Center for Latin American Studies at Harvard University.

### Competing Interests

The authors declare that they have no competing interests.

### Author Contributions

- Camila D. Ritter conceived and designed the experiments, performed the experiments, analyzed the data, contributed reagents/materials/analysis tools, prepared figures and/or tables, authored or reviewed drafts of the paper, approved the final draft.
- Alexander Zizka and Fabian Roger analyzed the data, contributed reagents/materials/-analysis tools, authored or reviewed drafts of the paper, approved the final draft.

- Hanna Tuomisto and Christopher Barnes authored or reviewed drafts of the paper, approved the final draft.
- R. Henrik Nilsson conceived and designed the experiments, analyzed the data, contributed reagents/materials/analysis tools, authored or reviewed drafts of the paper, approved the final draft.
- Alexandre Antonelli conceived and designed the experiments, contributed reagents/materials/analysis tools, authored or reviewed drafts of the paper, approved the final draft.

## Field Study Permissions

The following information was supplied relating to field study approvals (i.e., approving body and any reference numbers):

The sample collection was authorized by Brazilian authorities: ICMBio (registration number 48185-2) and IBAMA (registration number 127341).

## Data Availability

The raw data are provided in the Supplemental Files.

## Supplemental Information

Supplemental information for this article can be found online at http://dx.doi.org/10.7717/peerj.5661#supplemental-information.

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
