# Peer review of "High-throughput metabarcoding reveals the effect of physicochemical soil properties on soil and litter biodiversity and community turnover across Amazonia"

_PeerJ, doi:10.7717/peerj.5661_

## Round 0.1 · original submission · Major Revisions

This is a very well written manuscript based on the results from a very well designed experiment. Nonetheless, the referee´s raised a couple of good questions that will need to be addressed before being acceptable for publication in Peerj. Please, address those comments and all minor comments raised by the referee´s in a revised version of the manuscript.

The author´s should improve their hypotheses based on the questions and state-of-the art on this field. Results and discussion should be modified accordingly to connect hypotheses with findings and conclusions, avoiding unnecessary speculation. All that should be also well reflected in the abstract of the manuscript. Specifically, the author´s need to address well why they have chosen richness as the measure of diversity instead of other, more ecologically sound measures such as Shannon. In the same line, why they used presence/absence for their analyses instead of the abundance of each OUT, which gives more information. The referee´s also claims author´s should also improve their hypotheses and justifications on why to study litter and soil layers separately. I totally agree on separate them, but some work need to be done on justifying that decision. The referee #2 raised also several other methodological issues that will need to be carefully addressed.

The author´s should also take care when interpreting their results given that an important property of the soil/litter such as N content was not measured. Another important issue is the apparently artificial upscaling of the results to the whole Amazonian basin made by the author´s, which seems to be a bit exaggerated given the very discrete number of sampling sites compared with the spatial vastness and the huge biodiversity of the Amazon.

Reviewer 1 ·

Basic reporting

Justify why you are asking question one. Why do you expect a different between the litter and soil layers?

Experimental design

Why not use a Procrustes analysis to evaluate the dissimilarity patterns (as estimated by the Jaccard) between the litter and soil layers?

Validity of the findings

The sampling was minimal when compared to the vastness of the Brazilian Amazon, but no sampling is perfect. Better care should be taken when discussing the results in the larger context of this entire region.

Additional comments

Line 29: “In contrast” to “By contrast”.


Lines 70-71 and 329-331: Bates at al. (2013, ISME J. 7: 652-659) also found pH drives soil protist diversity.


Lines 154-150: State chemistry used in the Illumina sequencing. State primers and their references. Ritter et al. (in review) should not be used as a reference as it is not published.


Lines 164-165: Justify clustering similarity value used.


Line 331: “dominant group” to “dominant eukaryotic group”.


Lines 310-313: Some studies would disagree that fungi are the dominant tropical eukaryotic group.


Figure 5: Use different point shapes to distinguish the different habitats. When printed in black-and-white, the colors all look the same.

Reviewer 2 ·

Basic reporting

This paper describes a study on microbial prokaryotic and eukaryotic communities in soil and litter fractions, and their correlations with physico-chemical properties along the Amazon river in the Brazilian Amazonia. The authors use multivariate analyses and correlations between prokaryotic and eukaryotic OTU data tables and physical and chemical properties to investigate the main physico-chemical properties influencing the structure of these microbial communities in tropical forests.

The article is well written and well-structured with clear and unambiguous writing and professional English used throughout. The authors present clearly their objectives, methodology and results. Sufficient context is provided. However, I would suggest the authors stating clearly their hypotheses based on the previous research and adding some additional information in the methods that is notably missing.

All appropriate raw data is available (see comments on general comments to authors).

Figures are relevant to the content of the article, of good quality and appropriately described and labelled.

Experimental design

I consider that this article shows original primary research within the scope of PeerJ journal

The methodology is well described with sufficient detail to be reproducible by another researcher. It is appropriate for the questions showed in the paper. The research has been conducted to a high technical standard and in conformity with the prevailing ethical standards in the field. However, I consider that ‘soil’ and ‘litter’ should be appropriately described in section 2.1. There is a complete description of land uses and location of the plots, but it is not clear how the authors collected litter and soil substrates. Therefore, I strongly suggest describing this important sampling methodology.

Validity of the findings

The data are robust, statistically sound and controlled. Discussion is rather speculative as one important property (N content) is missing which would have improved the discussion of the manuscript towards the nutrient cycling issue. However, the paper shows important results that have been properly analysed and they worth its publication in PeerJ.

The conclusions lack some connection to the original questions investigated so I would suggest the authors re-writing this section including the conclusions linked to scientific questions.

Additional comments

I find that this paper is well written, well-structured and the methodology is appropriate for the questions raised by the authors. However, from my point of view, some important issues would need to be addressed before its publication (see comments below). Therefore, I suggest major revisions for its publication in PeerJ.

The research questions are well defined, but they still need to be well introduced to make them relevant and meaningful. The state of the art of the first question of the paper (differences between microbial communities’ structure and richness between soil and litter and their interactions with abiotic environmental properties) is not well defined as the main hypothesis regarding this question is not stated. What do the authors expect? This should be defined based on previous bibliography.

Additionally, important methodological information is missing: litter and soil description and sampling (which is particularly important for this paper). Therefore, I strongly recommend to describe properly what the authors understand by ´litter´ and ´soil´ fractions. What is the depth?, was it the same for the different land uses? Is ´litter´ the plant debris? Does it include woody/root debris?

Generally, the R script is clear and easy to follow but has some important issues that would need to be addressed, particularly in the second part of the script that cannot be run:
• Compilation to word document did not work so I followed the script.
• The ‘install.packages’ command is missing for the package called ‘splancs’
• The files provided have a different name to the filenames used in the script (e.g. 16S_all.csv). I suggest changing the names in the script (to e.g. peerj-26029-16S_all).
• A folder ‘output’ would need to be created to save output files but it is not described in the script.
• Line 117, not quite sure what you want to do here: OTUSbiol <- select(dat, Locality:OTUS_18SS), there is not any column with that name in ‘dat’ dataframe.
• Line 350: change ‘summary(s18l.nmds)’ to ‘summary(s18.nmds)’
• Line 438: Second hypothesis: I was not able to follow this part of the script as the ‘mice’ function is applied to dat, but the last reference to ‘dat’ is:
dat <- read.csv("peerj-26029-metadata.csv")
dat <- data.frame(dat[,1:7])
names(dat) <- capitalize(names(dat))
so it is not possible to apply mice as it is in the script: tempData <- mice(dat[1:39,8:27], m=5, maxit=50, meth='pmm')
• Hereafter, the script cannot be run (from line 438). So I strongly suggest the authors revising the script and running it again to be sure it can be run in other computers.

I suggest the authors reading the paper McMurdie PJ, Holmes S. 2014. Waste Not, Want Not: Why Rarefying Microbiome Data Is Inadmissible. PLOS Computational Biology 10(4): e1003531 for future research and I would suggest the authors not using rarefaction in the future and particularly for this paper as they are using presence/absence matrices to do their analysis. A priori, I would think rarefaction is not needed if working with presence/absence and with richness indexes. Another important question, why do the authors use presence/absence data? it should be justified in the text.

Discussion is well written and somewhat speculative in the nutrient cycling aspect. However, PeerJ welcomes speculative discussions if they are appropriately discussed as it is here. The nutrient cycling discussion would have been strongly supported if Nitrogen would have been measured to estimate C:N rates and discuss in depth the nutrient cycling and stoichiometry.

Some more detailed details are showed below:

Line 31: The aims of the work are notably missing in the abstract.

Line 76: Porazinska et al. 2012 showed differences in taxonomic composition for nematodes, it would be good to specify it.

Line 96: ‘Physio-chemical’ should be corrected to ‘physico-chemical’

Line 101: It is not clear what the authors expect from these data in contrast with the antecedent literature. I would suggest specifying the hypothesis for the first question.

Line 133: What do the authors mean by ‘determining the physicochemical and nutrient profiles of the first 5 cm’? Did they measure ‘along the profile from surface to 5 cm’? I understand samples were taken from the first 5 cm, so I suggest rewriting this sentence to ‘determining the physicochemical and nutrient properties of the first 5 cm’

Line 145: I suggest using ‘coarse’ instead of ‘thick’ when referring to sand (line 184 too).

Line 198: Apart from the ‘envfit’, another possible approach would have been to perform a Permutation Analysis of Variance (PERMANOVA test) to investigate if these factors do have a significant effect on the structure of the communities.

Line 206: I suggest changing ‘correlates’ to ‘correlated’.

Line 236: Here the authors are showing the adj. R2 whilst the R is showed in the figure caption. It is confusing, and I suggest showing only one value (I suggest adj. R2).

Line 247: This might be due to the rarefaction, it should be discuss in the discussion section.

Line 270: I suggest changing ‘soil properties’ to ‘soil/litter properties’.

Lines 279 and 280: Change ‘physic’ to ‘physico’

Line 302: A comma should be deleted in the reference Delgado-Baquerizo.

Line 351: I suggest changing ‘puzzling’ to ´unexpected´.

Line 370: I would suggest changing ‘diversity’ to ‘richness’.

Tables:

Table 1: Caption: Change OUT to OTU.
Table 1: ‘As the organic carbon content and pH are important variables to soil biota, we use them as independent variables.’ This is not explained in the text and should be specified in the Materials and Methods section.

Figures:

Figure 2. I suggest adding the regression fitness coefficients to the figures.
Fig. 3. I would delete ‘in Amazonian samples’ in the figure caption title as only Amazonian samples are showed in this paper.
Figure S4. In figure caption, I suggest changing ‘Blues circle are soils samples and red are litter samples. The sizes…’ to ‘Blue circles represent soil samples and red represent litter samples. The size…’.
There is not any figure S3, so I suggest numbering S4 as S3 and in the text.

·

Basic reporting

The report is well written.

Some references are missing (check attached file).

The Raw data (fastq files) are not publicly available, they should be submitted to a public database like SRA.

Experimental design

Successfully performed.

Validity of the findings

The main problem I have found in this study is that richness is assumed as diversity. The relative abundance of each OTU is not taken into account, nor is evenness taken into account when discussing the composition of the community.

Some assumptions may be wrong because of the problem discussed above.

Some data in the text are not supported in any table or figure (lines 307-311 and line 320).

As for the rest of the text, I find it an interesting and well-reported study.

Additional comments

I suggest that the authors conduct the study and comparisons taking into account the relative abundance of each OTU when comparing the composition of the different communities.

---

## Round 0.2 · Minor Revisions

After reading the suggestions from the 3 independent reviewers, I am happy to inform you that we will be glad to accept this manuscript for publication on Peerj. I have seen that the authors have thoroughly addressed all concerns and comments from the 3 reviewers.

Still, Authors should address some concerns from reviewer 3 (see enclosed comments).

Also, from my side, I am still a bit concerned with the low variability of the microbial richness/abundance explained by the physical/chemical variables. I see that physical/chemical variables only explain 30-35% of prokaryotic variability and 12-16% of fungal variability. This lack of predictability is, according to me, very valuable information, which should be acknowledge and discussed. We are unable to properly predict the diversity in tropical systems and this is something remarkable that should be, somehow, highlighted. Therefore, I encourage the authors to briefly acknowledge and discuss this in the manuscript.

Reviewer 1 ·

Basic reporting

Now good.

Experimental design

Now good.

Validity of the findings

Now good.

Additional comments

In this revision, I feel that authors have successfully and fully answered my previous comments.

Reviewer 2 ·

Basic reporting

The review of the article is well written and well-structured. The authors have considered all the comments from the reviewers, particularly the comments on the description of the state of the art, missing information in the methods section and new data analyses.

Experimental design

The authors have considered the comments from the reviewers and have improved the description of the methods, particularly providing a detailed description of ‘soil’ and ‘litter’ sampling in section 2.1.

Validity of the findings

The authors have reworded the conclusions section and they are now connected to the original questions investigated.

Additional comments

The research questions are well defined, and their introduction has been meaningfully improved. Methods are described with sufficient detail. Discussion is well written and has also been improved. Therefore, I would suggest accepting the paper for its publication in PeerJ.

One minor comment: It is mentioned that freezing soil samples was not feasible (line 168). If that was the case, how were samples stored/treated after soil sampling until DNA extraction? It would be good to specify this in methods section.

Some very minor details are showed below:
Abstract, line 47: I suggest modifying “…diversity of in one…” to “…diversity in one…”.
Lines 162-163: Organic Matter abbreviations are inconsistent between these two lines.
Table 2: Change “besence” to “absence”.

·

Basic reporting

The text is well written. It contains a few minor mistakes that have to be solved.

Experimental design

no comment

Validity of the findings

no comment

---

## Round 0.3 · Minor Revisions

I´ve seen that the author´s have properly addressed all our concern. So the manuscript is ready for publication in Peerj.
However, I still have some minor comments:
I see the author´s talk indistinctly about "differences" or "distances" along the text, especially in the discussion. Also is very confusing when they mix the concepts of "diversity", "community composition" and "turnover". Could you please fix this in order to be consistent?.
I also think that there is something wrong with some quotes to figures in the text (e.g. in line 441 do you refer to Figure 6 or to Table 1?)
In general, this new paragraph added to the discussion (lines 438-445) confuse more than clarifies!

---

## Round 0.4 · accepted · Accept

No further changes/comments required. This manuscript is ready for publication . I think it will be a great contribution to PeerJ.

Congratulations!

#